# Metal Contents in Fish from the Bay of Bengal and Potential Consumer Exposure—The EAF-Nansen Programme

**DOI:** 10.3390/foods10051147

**Published:** 2021-05-20

**Authors:** Amalie Moxness Reksten, Zillur Rahman, Marian Kjellevold, Esther Garrido Gamarro, Shakuntala H. Thilsted, Lauren M. Pincus, Inger Aakre, John Ryder, Sujeewa Ariyawansa, Anna Nordhagen, Anne-Katrine Lundebye

**Affiliations:** 1Seafood, Nutrition and Environmental State, Institute of Marine Research, P.O. Box 2029 Nordnes, 5817 Bergen, Norway; marian.kjellevold@hi.no (M.K.); inger.aakre@hi.no (I.A.); nordhagen_94@hotmail.com (A.N.); anne-katrine.lundebye@hi.no (A.-K.L.); 2Quality Control Laboratory, Department of Fisheries, Ministry of Fisheries & Livestock, Khulna 9000, Bangladesh; zrahmantec@gmail.com; 3Fisheries and Aquaculture Department, Food and Agriculture Organisation of the United Nations (FAO), 00153 Rome, Italy; esther.garridogamarro@fao.org (E.G.G.); john.ryder@fao.org (J.R.); 4WorldFish, Jalan Batu Maung, Batu Maung, Bayan Lepas 11960, Penang, Malaysia; s.thilsted@cgiar.org (S.H.T.); L.Pincus@cgiar.org (L.M.P.); 5National Aquatic Resources Research and Development Agency, Crow Island, Colombo 01500, Sri Lanka; sujeewa@nara.ac.lk

**Keywords:** arsenic, cadmium, mercury, lead, Bangladesh, Sri Lanka, fish, risk assessment

## Abstract

Fish represent an important part of the Sri Lankan and Bangladeshi diet. However, fish is also a source of contaminants that may constitute a health risk to consumers. The aim of this study was to analyse the contents of arsenic, cadmium, mercury, and lead in 24 commonly consumed marine fish species from the Bay of Bengal and to assess the potential health risk associated with their consumption. Mercury and lead contents did not exceed the maximum limits for any of the sampled species, and consumer exposure from estimated daily consumption was assessed to be minimal for adults and children. Numerous samples exceeded the maximum limit for cadmium (58%), particularly those of small size (≤25 cm). However, consumer exposure was insignificant, and health assessment showed no risk connected to consumption. These data represent an important contribution to future risk/benefit assessments related to the consumption of fish.

## 1. Introduction

Fish represent an important and rich source of essential and bioavailable nutrients, such as long-chain polyunsaturated fatty acids, protein, vitamin A, vitamin B_12_, vitamin D, calcium, iodine, and selenium [1,2]. We have previously shown that several of the marine fish species from Sri Lanka and Bangladesh may provide 25% or more of the recommended daily nutrient intake for multiple micronutrients [3,4]. In Sri Lanka, fish is estimated to contribute to approximately 55% of total animal protein intake per capita [5], whereas, in Bangladesh, the contribution is estimated to be around 60% [6]; thus, fish is the most important animal source of food in both countries [7]. However, fish may also be a source of various contaminants, such as metals, persistent organic pollutants (POPs), and plastics. Metal pollution of the aquatic environment may derive from anthropogenic activities, such as industrial processes and industrial waste, mining and smelting operations, and domestic and agricultural use of metals and metal-containing compounds, as well as a result of natural phenomena, such as volcanic eruptions and weathering [8,9,10,11]. As developing countries, both Sri Lanka and Bangladesh are at higher risk of metal pollution of the aquatic environment as a consequence of rapid urbanisation and industrialisation [12,13,14]. Heavy metals (such as cadmium, mercury, and lead) and metalloids (such as arsenic), hereafter combined and referred to as “metals”, readily bioaccumulate in the food chain and are characterised by a high level of human toxicity, even at low concentrations. Due to their long persistence in the environment, they rank among the top five priority metals that are of public health significance in the world today [11,15]. These metals are all classified as either “known” or “probable” human carcinogens, and exposure is associated with a host of negative health outcomes, including renal, skeletal, cardiovascular, and neurological damage, diminished cognitive function, developmental anomalies in children, and mortality at high concentrations [10,15,16].

To ensure harmonisation between the fisheries sector and various food safety standards, several guidelines, guidance documents, and regulations for risk management at national, regional, and international levels have been established. For example, The Codex Alimentarius Commission is a food standard setting body that constitutes a central part of the Joint Food and Agriculture Organisation (FAO)/World Health Organisation (WHO) Food Standards Programme that was established by FAO and WHO to protect consumers’ health and promote fair practices in food trade. Maximum exposure levels delineate a consumption threshold above which consumers are likely to be adversely exposed to various contaminants and are specific for each metal in various food products. Fish and fish products intended for export are subject to strict export regulations; in Sri Lanka and Bangladesh, local export regulations are based primarily on European Union (EU) legislation as the EU, Japan, and the United States of America (USA) are the three top destinations for fisheries exports from both countries [13,17,18].

Fish consumption is one of the major routes of arsenic, cadmium, mercury, and lead exposure to humans [19,20]. Fish concentrate mercury as the highly toxic form of organic methylmercury (MeHg), either transformed from inorganic mercury in the aquatic environment by anaerobic bacteria (methylation) or through feed components such as plankton and small fish in the aquatic food chain [20,21,22]. EU legislation has established maximum permissible levels for cadmium (0.050 mg/kg wet weight (w.w.)), mercury (0.5 mg/kg w.w.), and lead (0.3 mg/kg w.w.), but not for arsenic (Table 1) [23]. The maximum limit for lead in fish is in line with the guidelines provided by The Codex Alimentarius in its General Standard for Contaminants and Toxins in Food and Feed (CXS 193-1995). The Codex Alimentarius Commission has not established a maximum limit for mercury in fish; however, a level of 0.5 mg/kg w.w. has been established for MeHg, which is similar to the level established for total mercury by the EU [24]. For arsenic compounds in fish, no maximum content has yet been established. Fish is a major source of total arsenic exposure, but the majority of the arsenic is in the form of organic arsenic, specifically arsenobetaine, which is virtually nontoxic [19,21].

Because dietary intake of fish is a major mechanism of human exposure to these metals compared to other routes such as inhalation and dermal contact [15,18], regular monitoring of concentrations in fish and seafood and knowledge of seafood consumption are essential. The Joint FAO/WHO Expert Committee on Food Additives (JECFA) is an international expert committee administered jointly by the FAO and WHO that carries out risk assessments for food additives (intentionally added), natural toxins, and contaminants among others. The JECFA has established provisional tolerable weekly intake (PTWI) levels for contaminants, which sets limits for the maximum intake of contaminants in food that may be consumed weekly over a lifetime, without causing any adverse health effects. The term “tolerable” is used to signal permissibility rather than acceptability of the intake of contaminants inevitably associated with the intake of various foods, and the value is given on a weekly basis to allow for daily variations in intake. To express the cautious nature of the evaluation, the term “provisional” is used to signify the dynamic nature of the values; new data on the health impacts of human exposure to these metals may change the PTWI levels if deemed necessary by the committee [27]. In 2010, the JECFA recommended that expressing the tolerable intake of cadmium monthly would be more appropriate than indicating it weekly, considering the metal’s long half-life. Thus, the maximum safe cadmium intake is now set as a tolerable monthly intake of 25 µg/kg body weight (b.w.) [28]. For mercury, the JECFA has established two PTWI values: one applicable for dietary exposure from foods other than fish and seafood (inorganic mercury at 4 µg/kg b.w.) and the other for fish and seafood products (MeHg at 1.6 µg/kg b.w.) [29]. The PTWI for lead was revised from 0.05 mg/kg b.w. to 0.025 mg/kg b.w. in 1993; however, in 2010, the JECFA confirmed that lead exposure is associated with an increase in systolic blood pressure in adults and impaired neurodevelopment in children. Thus, it was concluded that the PTWI could no longer be considered health-protective and, therefore, the value was withdrawn. No new PTWI value for lead has been established [28]. Similarly, for arsenic, the previously established PTWI of 15 µg/kg b.w. was withdrawn in 2011 when the JECFA determined the lower limit of the benchmark dose for a 0.5% increased incidence of lung cancer (BMDL_0.5_) to be in the region of the PTWI value and, therefore, no longer appropriate. Since then, no new PTWI value for arsenic has been established [30].

The Bay of Bengal occupies the north-eastern end of the Indian Ocean and borders India to the west, Sri Lanka and Indonesia to the south, Myanmar to the east, and Bangladesh to the north. The water body is recognised as highly dynamic and ecologically diverse. Major sources of marine pollution in the Bay of Bengal include industrial, agrochemical, and municipal wastes, in addition to oil pollution. Furthermore, the many large rivers flowing into the Bay of Bengal, including the Ganges, the Mahanadi, and the Krishna, are substantial carriers of domestic and industrial waste into the coastal and marine waters in the Bay, as proper waste disposal facilities are lacking in the surrounding countries [13,31]. Because many fish species occupy high trophic levels in the food chain, they are considered good bioindicators of pollutants in the aquatic environment and represent a good monitoring tool to assess changes in the environment [32,33,34]. However, information on metal contamination in numerous marine fish species found in the Bay of Bengal is limited, and research is needed to ensure that fish consumption does not pose a risk to consumer health. The aim of this study was, therefore, to investigate the presence of arsenic, cadmium, mercury, and lead in commonly consumed marine fish species, as well as two mesopelagic fish species from Sri Lanka and Bangladesh, to ascertain whether the contents were compliant with the maximum limits defined by legislations. Furthermore, the potential human exposure and potential health risks of these fish species, based on consumption rates in Sri Lanka and Bangladesh, were assessed.

## 2. Materials and Methods

This paper uses data collected through scientific surveys with the research vessel (R/V) *Dr. Fridtjof Nansen* as part of the collaboration among the EAF-Nansen Programme, the National Aquatic Resources Research and Development Agency (NARA) in Sri Lanka, and the Department of Fisheries (DoF) in Bangladesh. The EAF-Nansen Programme is a partnership among the FAO, the Norwegian Agency for Development Cooperation (Norad), and the Institute of Marine Research (IMR), Bergen, Norway, for sustainable management of the fisheries of partner countries.

### 2.1. Sampling

Sampling of fish was conducted during surveys with R/V *Dr. Fridtjof Nansen* in the Bay of Bengal. The Sri Lanka survey was conducted from 24 June to 15 July 2018, and the Bangladesh survey was conducted from 3 to 15 August 2018. Sampling was carried out using pelagic (MultiPelt 624 trawl) and bottom trawls (Gisund Super bottom trawl), and the catch was subsequently sorted and identified according to species by taxonomists. Fish species were selected for sampling on the basis of their importance to the respective local diets as commonly consumed marine fish species, and the selection of species was advised by Sri Lankan and Bangladeshi marine and food scientists on board the vessel. Mesopelagic species were also included in the study due to the limited knowledge on the chemical composition of such species, even though they are currently not commonly consumed. The length (cm) of the fish from the tip of the head to the deepest fork of the caudal fin was measured, and the fish was weighed (g) on a marine measuring board. The fish were then categorised as either small fish (<25 cm) or large fish (>25 cm). Samples of fish were prepared according to the consumption style in local diets, as advised by the local scientists on board: as whole fish, including the skin, bones, and viscera, as fillets with skin and intramuscular bones, or as fillets only (excluding the skin, bones, and viscera of the fish) (Table 2). The categorisations of the fish (small or large) corresponded to the local eating practice of the fish sampled from Sri Lanka, where all small fish are commonly consumed whole, and only the fillets of large fish are commonly consumed. For the fish species sampled from Bangladesh, all samples of commonly consumed fish are eaten as fillets with skin and bones, and, as very few species were longer than 25 cm, all fish were categorised as “small”. The mesopelagic species were prepared whole, including the head, skin, tail, and viscera. Three composite samples of each species, consisting of five randomly selected individuals in each sample for large fish and a minimum of 20 randomly selected individuals for small fish (*n* was dependent on the total biomass of the species in order to obtain an adequate sample weight for the analyses), were prepared for contaminant analyses at the IMR. The samples were first minced and homogenised, using a food processor (Braun Multiquick 7 K3000, Kronberg im Taunus, Germany), and stored as wet samples at −20 °C in the freezer onboard the vessel. After a minimum of 12 h in the freezer, a subsample of each wet sample was freeze-dried for 72 h (24 h at −50 °C, immediately followed by 48 h at +25 °C, with a vacuum of 0.2–0.01 mbar, Labconco Freezone 18 L mod. 7750306, Kansas City, USA), and the dry matter was calculated on the basis of the weight change upon entering and exiting the freeze-dryer. Freeze-dried samples were then homogenised to fine powder using a knife mill (Retch Grindomix GM 200, Haan, Germany). The freeze-dried samples were vacuum-sealed and stored in insulated boxes in the vessel’s freezer at −20 °C until shipment by air cargo to the IMR laboratories, where the samples were stored at −80 °C, pending analyses.

### 2.2. Analytical Methods

Metal analyses were performed at the IMR laboratories. The laboratories regularly participate in national and international proficiency tests with satisfactory results to check the accuracy and precision of the analyses. The analyses were performed using accredited methods according to ISO 17025:2005, and Certified Reference Materials (CRM) were included in each sample run for quality control (CRM 1556b, oyster tissue, National Institute of Standards and Technology, Gaithersburg, USA and TORT-3, Lobster Hepatopancreas Reference Material for Trace Metals, National Research Council, Ottawa, ON, Canada). All values were within the accepted range of the analyses and gave a mean accuracy of 104% for arsenic, 99% for cadmium, 84% for mercury, and 82% for lead. The contents of arsenic, cadmium, mercury, and lead were determined in a total of 93 composite samples (three samples for each fish species) which were both homogenised and freeze-dried. Detailed information regarding the analytical methods performed, including limits of quantification (LOQ) and the measurement uncertainty for each metal, were described by Reksten et al. [35].

### 2.3. Data Management and Presentation of Analytical Data

The analytical data were exported from Laboratory Information Management System (LIMS) to Microsoft^®^ Office 365 Excel version 1910 for calculations of means and standard deviations (SD). All values are presented as means ± SD expressed in mg/kg w.w. of the three composite samples consisting of *n* individuals for each fish species. Statistical analyses were performed and graphs compiled using GraphPad Prism 8.3.0. The data did not meet the assumption of normality (tested using D’Agostino–Pearson normality test and Shapiro–Wilk normality test); thus, differences were considered significant by Mann–Whitney *t*-tests (nonparametric) when *p* < 0.05. Correlation analysis (Spearman) was used to compare the mean length of the species to the cadmium and mercury contents of each composite sample. For individual samples presenting analysed values <LOQ, a precautionary and conservative approach was used, assuming that the total amount of the contaminant present in the sample is equivalent to the LOQ value (upper bound LOQ). Thus, the unadjusted LOQ value was used when calculating the mean and SD of each species. For cadmium, four of 93 measurements were below the LOQ, whereas, for lead, 39 of 93 measurements (all samples of large species from Sri Lanka) were below the LOQ. For arsenic, no values were below the LOQ, whereas, for mercury, two of 93 measurements were below the LOQ. The content of mercury was measured as total mercury in this study; it can be assumed that 80–100% of the total mercury in fish is in the form of MeHg [26,30]. Similarly, the content of arsenic was measured as the total content of all arsenic compounds present in the samples.

### 2.4. Consumer Exposure

A preliminary estimation of the potential risk to human health related to cadmium and mercury exposure through fish consumption was established by evaluating the estimated daily intake of fish compared to the PTWI. This was determined on the basis of the average contents of metals in fish tissue and the average daily fish consumption rates. The values were presented as the percentage of the PTWI and PTMI values for MeHg and cadmium, respectively, as provided by the JECFA. For the exposure assessment of mercury, a precautionary approach was applied assuming that the total mercury in fish was entirely in the form of MeHg [36,37,38,39,40]. For cadmium, the values were calculated assuming a fish intake of four times per month (one time per week) for each fish species. The calculations were not performed for arsenic and lead as no PTWI values are available for these metals. The consumer exposure estimation was performed for both an average South Asian adult of 60 kg and a 10 year old child of 27 kg [41,42,43]. A daily Sri Lankan serving size of fish was estimated to be 43 g, according to the Household Income and Expenditure Survey (HIES) report from 2016 [44,45]. For Bangladesh, fish consumption was assumed to be higher at 54 g/day according to the HIES from 1991, 2000, and 2010 [46]. Due to the unavailability of intake rates for different age groups, an intake of 60% of that of an adult was assumed for children.

### 2.5. Health Risk Assessment

The methodology for estimation of noncarcinogenic risks and carcinogenic risks was applied in accordance with the methodology provided by the United States Environmental Protection Agency (US EPA) Region III’s Risk-Based Concentration Table [47]. For the calculations, we assumed that processing and cooking have no effect on the toxicity nor the contents of metals in fish, and the ingestion dose was assumed to be equal to the metal content.

#### 2.5.1. Noncarcinogenic Exposure

The noncarcinogenic risk for each metal through fish consumption was assessed using the target hazard quotient (THQ) [47], which is defined as the ratio between the measured content of the contaminant and the oral reference dose (RD), weighed by the length and frequency of exposure, amount ingested, and b.w.. The following equation was used (Equation (1)):THQ = {(EF × ED × FIR × C)/(RfD × BW × AT)} × 10^−3^,(1)
where THQ is the target hazard quotient for noncarcinogenic risks, EF is the exposure frequency (365 days/year), ED is the exposure duration in years (approximately 70 years for adults and 10 years for children), FIR is the food ingestion rate (43 g/day for Sri Lanka and 54 g/day for Bangladesh), C is the average metal content of each group of fish species (mg/kg w.w.), RfD is the estimated oral reference dose of the metal (mg/kg/day, 0.003 for arsenic, 0.001 for cadmium, 0.0005 for mercury, and 0.004 for lead) [48], BW is the average b.w. of an adult (60 kg) and a child (27 kg), and AT is the average exposure time for noncarcinogen effect (ED × 365 days/year). A THQ <1 implies that there is no adverse hazard to human health, whereas a THQ value >1 implies potential health risks associated with fish intake and, therefore, related interventions and protective measures should be taken [48].

To assess the overall potential for noncarcinogenic effects from multiple metals, a hazard index (HI) was formulated, according to the guidelines for health risk assessment of chemical mixtures by the US EPA [47,48]. The HI is expressed as the sum of the THQ, and the equation is as follows (Equation (2)):HI = ∑THQ.(2)

#### 2.5.2. Carcinogenic Exposure

For carcinogens, risk was estimated as the incremental probability of an individual to develop cancer over a lifetime exposure to potential carcinogens [47]. The target carcinogenic risk (TR) was calculated using the following equation (Equation (3)):TR = {(EF × ED × FIR × C × CSFo)/(BW × AT)} × 10^−3^,(3)
where CSFo is the oral carcinogenic slope factor (mg/kg b.w./day) from the Integrated Risk Information System database [49]: 1.5 (mg/kg b.w./day)^−1^ for arsenic, 0.38 (mg/kg b.w./day)^−1^ for cadmium, and 0.0085 (mg/kg b.w./day)^−1^ for lead. As no CSFo value is known for mercury, TR was not calculated for this metal.

## 3. Results

### 3.1. Sample Characteristics

This study included a total of 1111 individual samples of fish comprising 24 different species from the pelagic, mesopelagic, demersal, and reef zones of the Bay of Bengal, sampled from Sri Lankan and Bangladeshi marine waters. An overview of the identification details and the weight and length parameters of all species sampled are described in Table 3. The Sri Lankan fish species *Sphyraena jello* had the greatest mean weight of 2885 g and mean length of 88.5 cm.

### 3.2. Metal Contents in Fish Species from Sri Lanka

The arsenic, cadmium, mercury, and lead contents in the fish species from Sri Lanka, expressed on a wet weight basis, are listed in Table 4. In general, the metal contents varied widely among different fish species; the mean content of arsenic in all fish samples was the highest (1.9 mg/kg), followed by cadmium (0.19 mg/kg), and mercury (0.06 mg/kg), whereas that of lead was the lowest (0.01 mg/kg). Additionally, the contents varied between small and large species, indicating that the size of the fish and/or which tissues of the fish are consumed may be of importance. The difference between small and large fish was significant for arsenic (*p* = 0.0059), cadmium (*p* < 0.0001), lead (*p* < 0.0001), and mercury (*p* < 0.001). The highest content of arsenic, 9.27 mg/kg, was found in the small species *Decapterus macrosoma^1^*, followed by the large species *Diagramma pictum*, with 5.47 mg/kg. However, the other sample of *Decapterus macrosoma^2^*, sampled at a different location, contained a substantially lower arsenic content (0.83 mg/kg). The mean arsenic content for small species was generally higher than that of large species. The highest content of cadmium, 1.043 mg/kg, was also found in the same sample of *Decapterus macrosoma^1^*, which is more than 20 times above the maximum limit of 0.050 mg/kg set by the EU. Of the 12 small fish species analysed, 11 species (92%) presented a cadmium content in excess of the maximum limit. The mean cadmium content in small species was also substantially higher than that of large species, where none exceeded the maximum limit set by the EU. This relationship was also found to be significant by a negative correlation coefficient (*r* = −0.61, *p* < 0.0001). For mercury, the highest content of 0.347 mg/kg was found in the large species *Sphyraena jello*. Contrary to the other metals, the mean mercury content was significantly higher in large species compared to small species, as evidenced by a positive correlation coefficient for length and mercury content (*r* = 0.81, *p* < 0.0001). However, none of the sampled fish species from Sri Lanka exceeded the maximum value of 0.50 mg/kg for total mercury set by the EU. The highest content of lead, 0.081 mg/kg, was found in the small species *Leiognathus dussumieri;* a value that is far from the maximum limit of 0.30 mg/kg. Small species had a mean lead content approximately four times that of large species. Furthermore, a significant negative correlation between length and lead content was observed (*r* = −0.76, *p* < 0.0001).

### 3.3. Metal Contents in Fish Species from Bangladesh

The arsenic, cadmium, mercury, and lead contents in the sampled fish species from Bangladesh, expressed on a wet weight basis, are listed in Table 4. The mean metal contents in the fish species from Bangladesh followed a similar pattern to that of the fish species from Sri Lanka: highest for arsenic (1.8 mg/kg), followed by cadmium (0.082 mg/kg), mercury (0.017 mg/kg), and lead (0.023 mg/kg). The highest content of arsenic of 3.13 mg/kg was found in *Pentaprion longimanus*. For cadmium, seven of 12 species (58%) exceeded the EU maximum limit of 0.050 mg/kg, with the highest content (0.193 mg/kg) found in *Dussumieria elopsoides^2^*. However, the mean cadmium content in all fish species from Bangladesh was significantly lower than the mean of small species from Sri Lanka (0.082 and 0.291 mg/kg, respectively, *p* < 0.0001). The highest mercury content, 0.058 mg/kg, was found in *Megalaspis cordyla*; however, none of the samples exceeded the EU maximum value of 0.50 mg/kg. Overall, the mean mercury content in the samples from Bangladesh was similar to that of small species from Sri Lanka (0.017 and 0.015 mg/kg, respectively). For lead, the highest content, 0.062 mg/kg, was found in one sample of *Sardinella fimbriata*; however, no samples exceeded the maximum limit of 0.30 mg/kg. The mean lead content in the samples from Bangladesh was similar to that of small species from Sri Lanka (0.023 and 0.019 mg/kg, respectively). 

In terms of anatomical parts included in the analyses, the fish species from Bangladesh was most similar to that of large species from Sri Lanka (fillets with skin and intramuscular bones and fillets, respectively). When comparing the metal contents for these two groups (not including the two mesopelagic species from Bangladesh, as they were sampled whole), no significant differences were found for arsenic. However, the content of cadmium was significantly higher in the fish species from Bangladesh (*p* = 0.0001), whereas the content of mercury was significantly higher in the large species from Sri Lanka (*p* < 0.0001). For lead, the content was significantly higher in the fish species from Bangladesh (*p* = 0.0001).

### 3.4. Potential Consumer Exposure

Figure 1 shows how much one portion of the various fish species from Sri Lanka and Bangladesh contributes to the PTWI for MeHg as a percentage. The mean contribution from all small species from Sri Lanka was 1% for both adults and children, whereas large species contributed 6% and 8% for adults and children, respectively. The large species *Sphyraena jello* contributed most to the PTWI for both adults and children, with 16% and 21%, respectively. *Sphyraena jello* is the only species that, if one portion was consumed every day for 1 week, would exceed the PTWI for MeHg, with 112% for adults and 147% for children. The species from Bangladesh contributed very little to the PTWI of MeHg, with a mean contribution of only 1% for all species for both adults and children. *Megalaspis cordyla*, sampled from Bangladesh, had the highest contribution to MeHg: 3% for adults and 4% for children. Figure 2 illustrates how much one portion per week for 1 month of the various fish species from Sri Lanka and Bangladesh contributes to the PTMI of cadmium as a percentage. Contrary to the PTWI of MeHg, the mean contribution to cadmium intakes from small species from Sri Lanka (3% and 4% for adults and children, respectively) was higher than that of large species (1% and 2% for adults and children, respectively). Despite relatively low mean values for all sampled fish species, the small species *Decapterus macrosoma^1^* was identified to contribute substantially more than the other species, with 12% for adults and 16% for children. However, in order to exceed the PTMI for cadmium, a child would have to consume the species *Decapterus macrosoma^1^* every day for an entire month (120% of PTMI), whereas, for an adult, this consumption frequency would only account for 90% of the PTMI. Of the fish species from Bangladesh, *Dussumieria epolsoides^2^* was identified as the species with the highest contribution to the PTMI of cadmium, with 3% for adults and 4% for children. The mean contribution for all fish species from Bangladesh was 1% for adults and 2% for children.

### 3.5. Health Risk Assessment in Edible Tissues of Different Fish Species

The results of the health assessment of arsenic, cadmium, mercury, and lead regarding THQ, HI, and TR are presented in Table 5 for Sri Lanka and Bangladesh. THQ values for all metals were below the safe level of 1 for both adults and children, indicating that no adverse health effects are likely due to consumption of the sampled fish species with the present metal contents and consumption rates. When evaluating the effects from more than one metal, the HI was still below 1 for both adults and children, indicating no health risk. When evaluating the TR, the US EPA defines excess cancer risks as follows: a risk lower than approximately one chance in 1,000,000 (1 × 10^−6^) is considered negligible, whereas chances above 1 × 10^−4^ are sufficiently large to trigger some type of remediation [47]. Accordingly, cancer risks of less than 1 × 10^−4^ are generally considered acceptable. The carcinogenic risk levels associated with cadmium and lead for both children and adults were less than 1 × 10^−4^ for all species; this suggests that consuming these species is not associated with a high carcinogenic risk from exposure to cadmium or lead. However, the carcinogenic risk levels associated with arsenic for both adults and children exceeded the acceptable threshold level for all species, indicating potential cancer risks associated with consumption.

## 4. Discussion

This paper presented analytical data on the contents of arsenic, cadmium, mercury, and lead in 24 different marine fish species from the Bay of Bengal, sampled from Bangladeshi and Sri Lankan marine waters. All but two of these fish species are readily available and commonly consumed in the respective countries. These two mesopelagic species were included due to limited knowledge on the chemical composition of such species and their potential contribution as food and/or feed. Contents of mercury and lead were far below the EU maximum limits in fish for all species, whereas all but one of the small species from Sri Lanka and more than half of the species from Bangladesh exceeded the EU maximum limits for cadmium. However, the PTWI/PTMI for mercury and cadmium were not exceeded for any fish species on the basis of estimated consumption rates for adults and children. For total arsenic in fish, no maximum limit or PTWI value has been established, and, for lead, no PTWI value has been established. To the best of our knowledge, this is the first study within the scientific literature to report the contents of arsenic, cadmium, mercury, and lead in several of the analysed fish species from Sri Lanka: the small species *Encrasicholina devisi, Equulites elongatus, Leiognathus dussumieri*, and *Sillago ingenuua*; the large species *Lethrinus olivaceus* and *Nemipterus bipunctatus*; and from Bangladesh, the two mesopelagic species *Benthosema fibulatum* and *Bregmaceros mcclellandi*, and *Dussumieria elopsoides*.

The speciation of the arsenic compounds present in fish is the most important toxicological endpoint for human health assessments, and the total arsenic content, which was measured in this study, is not considered suitable for risk estimations [50,51]. However, some countries have specified a maximum limit for total arsenic in seafood. For example, in Hong Kong, the maximum limit is 6 mg/kg for fish and fish products, whereas in Australia and New Zealand, the value is 2 mg/kg for fish and crustacea [52]. According to these values, several of the sampled fish species in this study would have exceeded the maximum limits. In our study, the highest content of arsenic was reported in the small species *Decapterus macrosoma*, with 9.27 mg/kg. In a study where the arsenic content in fish fillet of cod, herring, mackerel, halibut, tusk, and saithe from the coast of Norway was analysed (*n* = 923), the authors reported that the content of total arsenic varied greatly between species, with values ranging from 0.3 to 110 mg/kg [53]. This is an extremely large range compared to the range of total arsenic reported in this study (0.09–9.27 mg/kg). In a Belgian market study, Ruttens et al. (2012) reported a mean range of 1.05–9.36 mg/kg in nine species of marine fish [54], which is much more similar to the results of this study. A similar range was also reported in a study from the coastal areas of Bangladesh; Raknuzzaman et al. (2016) reported contents ranging from 0.76–13 mg/kg in fresh fish. Furthermore, the authors also reported that the TR for adults was exceeded for arsenic with consumption of the sampled fish and crustacea from the Bay of Bengal [55]. In this study, the TR for both adults and children was also exceeded for arsenic in fish from Sri Lanka and Bangladesh. However, in this study, the content of total arsenic was analysed. The content of inorganic arsenic in fish, which is the toxic form, has in other studies been found to be very low, usually <1% of the total arsenic content and often below quantifiable amounts [51,53,54,56]. Thus, analyses identifying the speciation of the arsenic compounds in the sampled fish species from Sri Lanka and Bangladesh are recommended to further evaluate the potential carcinogenic health risks associated with consumption. Nevertheless, as high consumers of rice, the Sri Lankan and Bangladeshi people may be at risk of exposure to inorganic arsenic at hazardous levels through rice consumption, as inorganic arsenic accounts for a large proportion of the total arsenic in rice. Additionally, elevated contents of inorganic arsenic in the groundwater (and, consequently, drinking water) present a major problem in both countries, particularly in Bangladesh [52,57].

The mean cadmium content in small fish from Sri Lanka significantly exceeded that of both large fish from Sri Lanka and all fish from Bangladesh (0.291, 0.010, and 0.082, respectively). In a scientific report by the European Food Safety Authority (EFSA) assessing the cadmium content in fish from 20 member states of the EU (*n* = 6393), the mean content reported was 0.0137 mg/kg, which is in line with the mean values presented in this study for large fish from Sri Lanka and fish from Bangladesh. However, this value is considerably lower than the cadmium content in the small fish species from Sri Lanka. The maximum value reported in the report was 0.3000 mg/kg, a value that is closer to that of small species from Sri Lanka (0.291 mg/kg) [58]. In general, the cadmium content in the large fish species from Sri Lanka was similar to that of other large fish species sampled in the same area of the Bay of Bengal: yellowfin tuna (0.01 mg/kg), swordfish (0.09 mg/kg), black marlin (0.02 mg/kg), and red snapper (0.01 mg/kg) [59]. Of the small species sampled from Sri Lanka, considerably lower cadmium contents been reported in the literature (predominantly from other Asian countries): 0.06 mg/kg [60] and 0.03 mg/kg [61] for *Auxis Thazard* (0.18 mg/kg in this study), 0.04 mg/kg [60] and 0.008 mg/kg [61] for *Decapterus macrosoma* (mean value of 0.76 mg/kg in this study), 0.27 mg/kg [61] for *Stolephorus indicus* (0.51 mg/kg in this study), and 0.03 mg/kg for *Rastrelliger kanagurta* (0.27 mg/kg in this study) [62]. However, cadmium accumulation in fish is not homogeneous. Cadmium accumulates primarily in the kidneys, followed by the liver and gills of fish [63,64]. This may account for the significant differences observed between large and small fish species from Sri Lanka, as the head and viscera were excluded from the analyses of large fish and in most of the fish species from Bangladesh. Nevertheless, the two mesopelagic species from Bangladesh were both sampled whole and presented great differences in cadmium content; *Benthosema fibulatum* contained a more than five times higher content of cadmium than *Bregmaceros mcclellandi*, which had quite low contents of cadmium compared to the other species.

Over the last 10 years, the European Union’s Rapid Alert System for Food and Feed (RASFF) reported four and 69 notifications for Bangladeshi and Sri Lankan exports of fish and fish products, respectively. Of the four cases from Bangladesh, none were related to metal contamination, whereas, of the 27 cases related to metal contamination in Sri Lanka, 26 were related to mercury contents that exceeded the EU maximum limit of 0.50 mg/kg. The 26 cases were all for large fish species such as swordfish, tuna, and barracuda [65]. Generally, higher contents are observed in older and larger fish across and within species, as the mercury content is related to the age of the fish (as the fish have had longer time to accumulate mercury) and the position of the fish species within the food chain (bioaccumulation and biomagnification) [26]. The one species of barracuda (*Sphyraena jello*) included in this study contained a considerably higher content of mercury compared to all the other fish species (0.348 mg/kg) and was also substantially larger than the other species (88.5 cm). In a review article on mercury content in fish from Sri Lanka, Jinadasa et al. (2019) reported that most fish species were below the maximum limit, except for certain top-trophic-level fish species (swordfish, tuna, and marlin) [13]. In this study, *Sphyraena jello* was the only species to exceed the PTWI for MeHg if consumed every day by adults and children for a week. However, this is a worst-case scenario, and it is unlikely that an individual’s entire weekly fish intake consists of a single species with the highest content of mercury. In both Sri Lanka and Bangladesh, the consumption of small fish is more frequent in the diets of the poor, whereas households of higher socioeconomic status purchase larger fish species and often consume only the fillets [66,67]. Therefore, according to the results of this study, the poor, as high consumers of small fish species, may be at a lower risk of exposure to high contents of MeHg.

None of the analysed species in this study presented values close to the maximum limit of lead; 42% of the samples were below the LOQ, indicating that most species, especially large species from Sri Lanka, are not significant sources of lead exposure. Lead is mostly accumulated in the gills, liver, kidneys, and bones of fish [68]. This may explain the significant difference between large and small fish species sampled from Sri Lanka in this study, as these anatomical parts were excluded in the samples of large fish, thus possibly resulting in a lower lead content.

The metal bioaccumulation in fish is influenced by biotic and abiotic factors, such as the habitat and trophic level, geographic region, the temperature, salinity, and pH value of the surrounding water, as well as the age, gender, body mass, and diet of the fish [64]. This may explain the large differences in both arsenic and cadmium content between the two samples of *Decapterus macrosoma* from Sri Lanka (9.27 and 0.83 mg arsenic per kg and 1.04 and 0.47 mg cadmium per kg, respectively). This difference was not seen in any of the other samples from Bangladesh that were also sampled multiple times from different locations. In this paper, we reported significant differences between the length of the sampled fish species and their metal content. A possible explanation for the negative relationship between length and metal content for arsenic, cadmium, and lead is the difference in metabolic activity between younger and older fish, where age often corresponds to the size of the fish. The metabolic activity of younger individuals is higher than that of older individuals, and metal accumulation increases with higher metabolic activity; thus, younger individuals often have higher contents [69]. However, this dilution of tissue metal content due to growth and lowered metabolic activity is not seen for mercury, where a positive relationship exists between length and metal content due to biomagnification [26].

In this paper, we estimated the potential consumer exposure for adults and children on the basis of the average fish intake in Sri Lanka and Bangladesh as calculated from the respective HIES, due to the lack of comprehensive national dietary data and nutrition studies. However, a number of reliability issues are related to the use of such data (e.g., the lack of information on intra-household food distribution and the reliability of surveys employing interview methods in general) [70]. In addition, assumptions that the intake is higher for some individuals of the population must be assumed but were not accounted for in this paper. According to our data, the Sri Lankan and Bangladeshi populations are likely at minimal risk of metal exposure at a hazardous level from consumption of the fish species included in this paper. However, other seafoods, such as crustaceans, molluscs, and cephalopods, are also major sources of metals and an important part of the Sri Lankan and Bangladeshi diet [51,58,68,71]. Due to the omission of other seafoods when assessing the potential consumer exposure, we cannot exclude that the potential consumer exposure might be at a higher level than that assessed in this study. However, fish and seafood provide millions of people around the world with a vast variety of essential nutrients and represent a cheap and easily available means of nutritional diversification for people in many low- and middle-income countries, such as Sri Lanka and Bangladesh, that depend heavily on a narrow range of staple foods [5]. When evaluating the benefits and risks of fish and seafood consumption, EFSA concluded that the benefits (reduced cardiovascular disease in adults and improved functional neurodevelopment in children with fish consumption during pregnancy) of fish and seafood consumption in the range of 1–4 servings per week generally outweigh the potential risks (with some exceptions for species with high contents of mercury) [26]. This was also the conclusion in the Expert Opinion by the FAO/WHO in 2010 [72]. Although food safety issues are a continuous concern, reaching the Sustainable Development Goals (SDGs) by 2030 will not be feasible if fish and seafood are not part of key strategies, as fish and seafood play a key role in human nutrition and health, as well as the economic, social, and environmental sustainability of food systems [5].

## 5. Conclusions

In this paper, we presented analytical data on the contents of arsenic, cadmium, lead, and mercury in a large variety of marine fish species from the Bay of Bengal, several of which have not been analysed before. Our findings showed that the contents of mercury and lead in all of the sampled fish species did not exceed the EU maximum limits, and that the exposure to these metals from estimated daily fish consumption of the analysed fish species are minimal for adults and children. However, in several fish species, particularly the small species, the EU maximum limit for cadmium was exceeded. Nevertheless, the potential consumer exposure for cadmium was considered insignificant for both adults and children, given the estimated mean fish consumption rates in Sri Lanka and Bangladesh. Significant differences between small and large fish from Sri Lanka were found for all metals; small species had significantly higher contents of arsenic, cadmium, and lead, whereas large species presented a significantly higher content of mercury. The data presented in this study suggest that the sampled fish species pose no health risks to adults and children when consumed at estimated consumption rates and represent an important contribution to future risk/benefit assessments.

## Figures and Tables

**Figure 1 foods-10-01147-f001:**
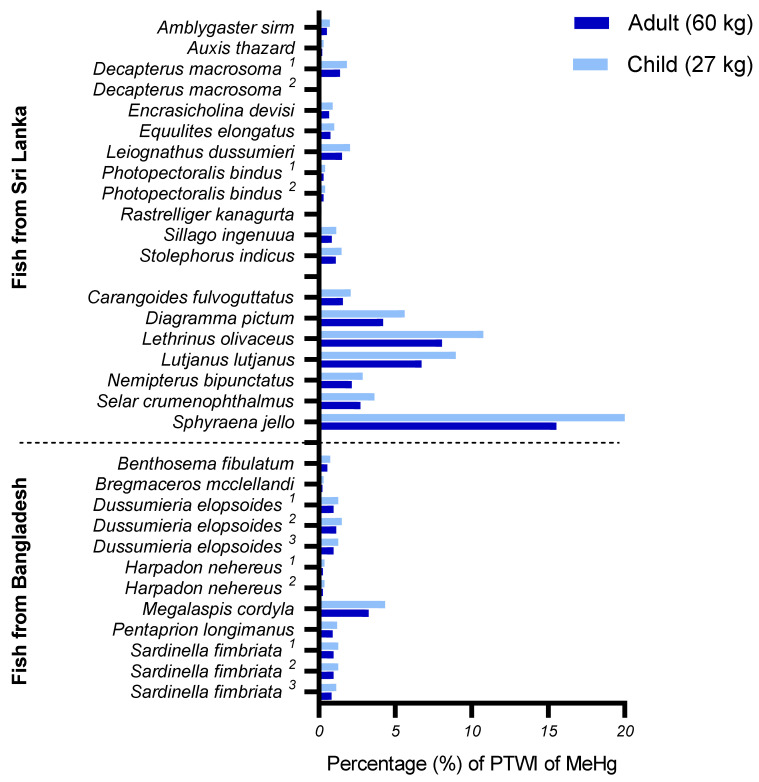
Potential consumer exposure for adults and children to methylmercury (MeHg), expressed as the metal content in one portion (43 and 26 g for Sri Lankan adults and children, respectively, and 54 and 32 g for Bangladeshi adults and children, respectively) of the various fish species from Sri Lanka and Bangladesh compared to the provisional weekly intake (PTWI) set by the JECFA. ^1,2,3^ Fish species sampled multiple times at separate locations.

**Figure 2 foods-10-01147-f002:**
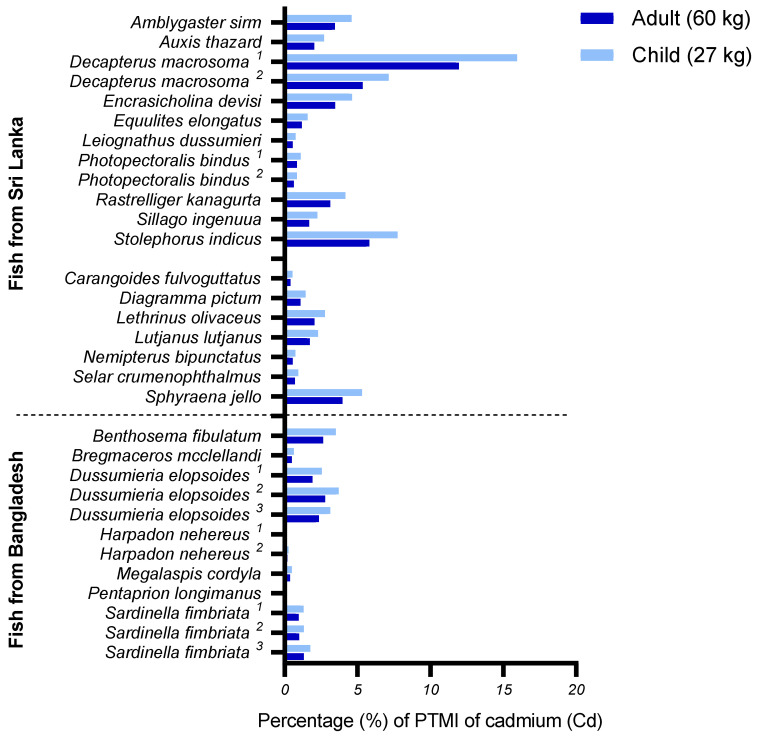
Potential consumer exposure for adults and children to cadmium (Cd), expressed as the metal content in one portion (43 and 26 g for Sri Lankan adults and children, respectively, and 54 and 32 g for Bangladesh adults and children, respectively) of the various fish species from Sri Lanka and Bangladesh compared to the provisional monthly intake (PTMI) set by the JECFA. For the calculations, a fish intake of four times per month (one time per week) for each fish species was assumed. ^1,2,3^ Fish species sampled multiple times at separate locations.

**Table 1 foods-10-01147-t001:** Maximum limits and tolerable weekly intakes for contaminants in muscle fillet of fish for arsenic (As), cadmium (Cd), mercury (Hg), and lead (Pb).

Metal	Maximum Limit (mg/kg w.w.)	Provisional Tolerable Weekly Intake (PTWI)
Arsenic (As)	- ^a^	- ^b^
Cadmium (Cd)	0.050 ^c^	PTMI: 25 µg/kg b.w. ^d^
Mercury (Hg)	0.50 ^c,e,f^	1.6 µg/kg b.w. ^g^
Lead (Pb)	0.30 ^c,e^	- ^b^

^a^ No maximum limit established for arsenic in fish or other seafood. ^b^ PTWI withdrawn by the Joint FAO/WHO Expert Committee on Food Additives (JECFA) due to no longer being considered health-protective. ^c^ European Union, Commission Regulation (EC) 488/2014 [25]. The EU maximum limit for cadmium is 0.050 mg/kg for most species, but is 0.1, 0.15, and 0.25 mg/kg for certain species. ^d^ Due to the long half-life of cadmium, the JECFA determined that indicating the tolerable intake monthly is more appropriate than weekly; thus, the PTMI value is presented. ^e^ The Codex Alimentarius Commission [24]. For mercury, the maximum limit provided by The Codex Alimentarius is for methylmercury (MeHg). ^f^ Certain predatory fish species of high trophic levels are excluded and possess a higher maximum limit of 1 mg/kg muscle [23,24]. ^g^ The PTWI is given for MeHg, as this is the form most commonly found in fish and seafood [20,21,22,26]. Abbreviations: b.w.: body weight; JECFA: Joint FAO/WHO Expert Committee on Food Additives; PTMI: provisional tolerable monthly intake; w.w.: wet weight.

**Table 2 foods-10-01147-t002:** Overview of species sampled, tissue, number of composite samples, and number of fish in each composite sample.

Scientific Name	Tissue Sampled	Number of Composite Samples	Number of Fish in Each Composite Sample
**Fish from Sri Lanka**			
**Small fish**			
*Amblygaster sirm*	Whole fish	3	25
*Auxis thazard*	Whole fish	3	25
*Decapterus macrosoma* ^1^	Whole fish	3	25
*Decapterus macrosoma* ^2^	Whole fish	3	25
*Encrasicholina devisi*	Whole fish	3	50
*Equulites elongates*	Whole fish	3	25
*Leiognathus dussumieri*	Whole fish	3	25
*Photopectoralis bindus* ^1^	Whole fish	3	25
*Photopectoralis bindus* ^2^	Whole fish	3	25
*Rastrelliger kanagurta*	Whole fish	3	25
*Sillago ingenuua*	Whole fish	3	25
*Stolephorus indicus*	Whole fish	3	25
**Large fish**			
*Carangoides fulvoguttatus*	Fillet	3	5
*Diagramma pictum*	Fillet	3	5
*Lethrinus olivaceus*	Fillet	3	5
*Lutjanus lutjanus*	Fillet	3	5
*Nemipterus bipunctatus*	Fillet	3	5
*Selar crumenophthalmus*	Fillet	3	5
*Sphyraena jello*	Fillet	3	5
**Fish from Bangladesh ^a^**			
*Benthosema fibulatum* *Bregmaceros mcclellandi*	Whole fish Whole fish	3 3	250 280
*Dussumieria elopsoides* ^1^	Fillet with skin and bones	3	25
*Dussumieria elopsoides* ^2^	Fillet with skin and bones	3	25
*Dussumieria elopsoides* ^3^	Fillet with skin and bones	3	23 ^b^
*Harpadon nehereus* ^1^	Fillet with skin and bones	3	25
*Harpadon nehereus* ^2^	Fillet with skin and bones	3	20
*Megalaspis cordyla*	Fillet with skin and bones	3	5
*Pentaprion longimanus*	Fillet with skin and bones	3	25
*Sardinella fimbriata* ^1^	Fillet with skin and bones	3	23 ^b^
*Sardinella fimbriata* ^2^	Fillet with skin and bones	3	25
*Sardinella fimbriata* ^3^	Fillet with skin and bones	3	25

^1,2,3^ Fish species sampled multiple times at separate locations. ^a^ Tissue consists of fillets with skin and small intramuscular bones. ^b^ One composite sample consisted of 22 fish, whereas the other two consisted of 23 fish.

**Table 3 foods-10-01147-t003:** Identification details and physical parameters of fish species sampled from Sri Lanka and Bangladesh ^a^.

Scientific Name	Common Name	Local Name		Habitat	Average Weight (g) ^b^	Average Length (cm)
**Fish from Sri Lanka**		**Sinhalese Name ^c^**	**Tamil Name ^c^**			
**Small fish**						
*Amblygaster sirm*	Trenched sardinella	Hurulla	Keerimeen saalai	Pelagic	278 ± 20	10.5
*Auxis thazard*	Frigate tuna	Alagoduwa	Urulan soorai	Pelagic	1180 ± 27	16.2
*Decapterus macrosoma* ^1^	Shortfin scad	Linna	Mundakan kilichchi	Pelagic	763 ± 23	13.5
*Decapterus macrosoma* ^2^	Shortfin scad	Linna	Mundakan kilichchi	Pelagic	273 ± 22	9.2
*Encrasicholina devisi*	Devis’ anchovy	Halmessa	Neththili	Pelagic	219 ± 1	10.5
*Equulites elongatus*	Slender ponyfish	Karalla	Karal	Demersal	183 ± 8	7.7
*Leiognathus dussumieri*	Dussumier’s ponyfish	Karalla	Vari karai	Demersal	637 ± 56	10.6
*Photopectoralis bindus* ^1^	Orangefin ponyfish	Karalla	Tatnam-kare	Demersal	245 ± 20	7.4
*Photopectoralis bindus* ^2^	Orangefin ponyfish	Karalla	Tatnam-kare	Demersal	228 ± 10	7.5
*Rastrelliger kanagurta*	Indian mackerel	Kumbalava	Kanang keluththi	Pelagic	610 ± 6	12.5
*Sillago ingenuua*	Bay whiting	- ^d^	Kelangan	Demersal	1099 ± 24	16.3
*Stolephorus indicus*	Indian anchovy	Halmassa	Neththili	Pelagic	676 ± 10	13.2
**Large fish**						
*Carangoides fulvoguttatus* ^e^	Yellowspotted trevally	Thumba parawa	Manjal parai	Reef-associated	168 ± 31	20.5 ± 1.5
*Diagramma pictum*	Painted sweetlips	Gobaya	Kallu kallewa	Reef-associated	1694 ± 906	47.9 ± 7.5
*Lethrinus olivaceus*	Long-face emperor	Uru hota	Thinan	Reef-associated	1886 ± 2275	46.4 ± 17.4
*Lutjanus lutjanus*	Bigeye snapper	Hunu ranna	Nooleni	Demersal	317 ± 58	27.5 ± 1.8
*Nemipterus bipunctatus* ^e^	Delagoa threadfin bream	- ^d^	Cundil	Demersal	78 ± 45	16.3 ± 3.2
*Selar crumenophthalmus* ^e^	Bigeye scad	Bolla	Chooparai	Reef-associated	174 ± 45	21.3 ± 1.7
*Sphyraena jello*	Pickhandle barracuda	Silava	Jeela	Reef-associated	2885 ± 557	88.5 ± 5.6
**Fish from Bangladesh**						
*Benthosema fibulatum*	Spinycheek lanternfish	Puiya		Mesopelagic	0.6	<5
*Bregmaceros mcclellandi*	Unicorn cod	- ^d^		Mesopelagic	0.5	<6
*Dussumieria elopsoides* ^1^	Slender rainbow sardine	Maricha		Pelagic	49.7	17.0
*Dussumieria elopsoides* ^2^	Slender rainbow sardine	Maricha		Pelagic	72.6	20.3
*Dussumieria elopsoides* ^3^	Slender rainbow sardine	Maricha		Pelagic	67.8	19.3
*Harpadon nehereus* ^1^	Bombay duck	Loittya		Demersal	110.3	25.5
*Harpadon nehereus* ^2^	Bombay duck	Loittya		Demersal	117.7	24.2
*Megalaspis cordyla*	Torpedo scad	Kuawa		Pelagic	114.6	25.2
*Pentaprion longimanus*	Longfin mojarra	Dom Mach		Demersal	20.8	11.4
*Sardinella fimbriata* ^1^	Fringescale sardinella	Chapila		Pelagic	35.2	16.1
*Sardinella fimbriata* ^2^	Fringescale sardinella	Chapila		Pelagic	40.3	16.4
*Sardinella fimbriata* ^3^	Fringescale sardinella	Chapila		Pelagic	43.5	16.4

^a^ Values are presented as means ± standard deviations (SD) and are based on length and weight values (prior to any handling) of the sampled fish species. ^b^ Weight measurements are expressed as the total weight of the composite sample consisting of *n* number of fish for small species, and per individual fish for large species from Sri Lanka (see Table 2). The length of small fish species from Sri Lanka and all fish species from Bangladesh was calculated as a mean value of the first composite sample measured as a whole during the surveys; thus, no SD is presented. ^c^ Only applicable for fish species from Sri Lanka. ^d^ The local names of all species were not available. ^e^ Species categorised as large fish (although their length was <25 cm) based on input on the eating practice of the species by the national scientists on board, for which only the fillet is commonly consumed (thus corresponding to the local eating practice of large fish and not small fish). ^1,2,3^ Fish species sampled multiple times at separate locations.

**Table 4 foods-10-01147-t004:** Contents of arsenic (As), cadmium (Cd), mercury (Hg), and lead (Pb) in fish species from Sri Lanka and Bangladesh (mean ± SD).

Species ^a^	As (mg/kg w.w.)	Cd (mg/kg w.w.)	Hg (mg/kg w.w.)	Pb (mg/kg w.w.)
**Fish from Sri Lanka**				
**Small fish**				
*Amblygaster sirm*	1.13 ± 0.06	0.300 ± 0.026 ^b^	0.012 ± 0.002	0.011 ± 0.001
*Auxis thazard*	1.10 ± 0.00	0.177 ± 0.021 ^c^	0.005 ± 0.001	0.006 ± 0.000
*Decapterus macrosoma* ^1^	9.27 ± 1.02	1.043 ± 0.100 ^b^	0.031 ± 0.005	0.018 ± 0.001
*Decapterus macrosoma* ^2^	0.83 ± 0.05	0.467 ± 0.035 ^b^	0.002 ± 0.000	0.006 ± 0.001
*Encrasicholina devisi*	0.93 ± 0.04	0.303 ± 0.023 ^b^	0.015 ± 0.001	0.009 ± 0.001
*Equulites elongatus*	1.80 ± 0.10	0.103 ± 0.006 ^b^	0.017 ± 0.001	0.012 ± 0.002
*Leiognathus dussumieri*	3.77 ± 0.38	0.049 ± 0.002	0.034 ± 0.003	0.081 ± 0.017
*Photopectoralis bindus* ^1^	1.57 ± 0.06	0.072 ± 0.040 ^b^	0.007 ± 0.001	0.031 ± 0.001
*Photopectoralis bindus* ^2^	1.90 ± 0.17	0.055 ± 0.005 ^b^	0.007 ± 0.007	0.020 ± 0.003
*Rastrelliger kanagurta*	0.69 ± 0.03	0.273 ± 0.006 ^b^	0.003 ± 0.001	0.009 ± 0.001
*Sillago ingenuua*	1.33 ± 0.12	0.147 ± 0.006 ^b^	0.019 ± 0.004	0.021 ± 0.007
*Stolephorus indicus*	1.80 ± 0.10	0.507 ± 0.020 ^b^	0.025 ± 0.002	0.008 ± 0.001
**Mean for small fish**	2.18 ± 2.32	0.291 ± 0.275	0.015 ± 0.011	0.019 ± 0.021
**Large fish**				
*Carangoides fulvoguttatus*	1.50 ± 0.30	0.002 ± 0.001	0.035 ± 0.004	0.005 ± 0.001
*Diagramma pictum*	5.47 ± 0.06	0.005 ± 0.001	0.094 ± 0.016	0.005 ± 0.000
*Lethrinus olivaceus*	1.00 ± 0.36	0.001 ± 0.000	0.180 ± 0.156	0.005 ± 0.000
*Lutjanus lutjanus*	0.69 ± 0.15	0.019 ± 0.004	0.150 ± 0.026	0.005 ± 0.001
*Nemipterus bipunctatus*	0.58 ± 0.09	0.012 ± 0.006	0.048 ± 0.015	0.005 ± 0.001
*Selar crumenophthalmus*	0.86 ± 0.09	0.027 ± 0.018	0.061 ± 0.022	0.006 ± 0.000
*Sphyraena jello*	0.64 ± 0.28	0.005 ± 0.002	0.347 ± 0.032	0.005 ± 0.000
**Mean for large fish**	1.53 ± 1.69 *	0.010 ± 0.011 ***	0.131 ± 0.116 **	0.005 ± 0.001 ***
**Fish from Bangladesh**				
*Benthosema fibulatum*	2.33 ± 0.06	0.183 ± 0.001 ^b^	0.010 ± 0.001	0.032 ± 0.001
*Bregmaceros mcclellandi*	1.23 ± 0.21	0.033 ± 0.002	0.004 ± 0.000	0.025 ± 0.015
*Dussumieria elopsoides* ^1^	1.77 ± 0.15	0.133 ± 0.032 ^b^	0.017 ± 0.003	0.007 ± 0.002
*Dussumieria elopsoides* ^2^	1.04 ± 0.14	0.193 ± 0.021 ^b^	0.020 ± 0.000	0.006 ± 0.001
*Dussumieria elopsoides* ^3^	1.37 ± 0.06	0.163 ± 0.006 ^b^	0.017 ± 0.001	0.006 ± 0.000
*Harpadon nehereus* ^1^	0.09 ± 0.01	0.007 ± 0.000	0.005 ± 0.000	0.020 ± 0.000
*Harpadon nehereus* ^2^	0.18 ± 0.04	0.013 ± 0.013	0.005 ± 0.001	0.020 ± 0.000
*Megalaspis cordyla*	2.23 ± 0.23	0.025 ± 0.002	0.058 ± 0.009	0.008 ± 0.004
*Pentaprion longimanus*	3.13 ± 0.35	0.005 ± 0.002	0.016 ± 0.001	0.012 ± 0.003
*Sardinella fimbriata* ^1^	3.07 ± 0.06	0.067 ± 0.008 ^b^	0.017 ± 0.001	0.048 ± 0.003
*Sardinella fimbriata* ^2^	2.87 ± 0.06	0.069 ± 0.005 ^b^	0.017 ± 0.000	0.035 ± 0.005
*Sardinella fimbriata* ^3^	2.23 ± 0.06	0.091 ± 0.006 ^b^	0.015 ± 0.002	0.062 ± 0.008
**Mean for all fish**	1.80 ± 1.01	0.082 ± 0.069 ***	0.017 ± 0.014 ***	0.023 ± 0.018 ***

^a^ The analytical value for each fish species is the mean of three composite samples, consisting of *n* number of samples (see Table 2) ^b^ Cadmium content exceeded the EU maximum limit of 0.050 mg/kg in muscle of fish. ^c^ Cadmium content exceeded the EU maximum limit of 0.15 mg/kg in muscle of *Auxis spp*. ^1,2,3^ Fish species sampled multiple times at separate locations. * *p* ≤ 0.01. Significant differences in metal contents when comparing the means of small and large fish species as a group for samples from Sri Lanka and the means of all fish species from Bangladesh to that of all fish species from Sri Lanka. ** *p* ≤ 0.001. Significant differences in metal contents when comparing the means of small and large fish species as a group for samples from Sri Lanka and the means of all fish species from Bangladesh to that of all fish species from Sri Lanka. *** *p* ≤ 0.0001. Significant differences in metal contents when comparing the means of small and large fish species as a group for samples from Sri Lanka and the means of all fish species from Bangladesh to that of all fish species from Sri Lanka. Abbreviations: SD: standard deviation, w.w.: wet weight.

**Table 5 foods-10-01147-t005:** Target hazard quotient (THQ), hazard index (HI), and target carcinogenic risk (TR) for arsenic, cadmium, mercury, and lead, based on estimated consumption rates of the fish species sampled from Sri Lanka and Bangladesh for adults (60 kg) and children (27 kg).

Metal	Mean Metal Content (mg/kg w.w.)	THQ		HI		TR	
**Fish from Sri Lanka**							
	**Small Fish**	**Adults**	**Children**	**Adults**	**Children**	**Adults**	**Children**
*As*	2.18	0.5208	0.6997	0.5984	0.8040	2.3 × 10^−3^	3.1 × 10^−3^
*Cd*	0.291	0.0695	0.0934			7.9 × 10^−5^	1.1 × 10^−4^
*Hg*	0.015	0.0036	0.0048			NA ^a^	NA ^a^
*Pb*	0.019	0.0045	0.0060			1.2 × 10^−7^	1.6 × 10^−7^
	**Large fish**						
*As*	1.530	0.3655	0.4911	0.4004	0.5379	1.6 × 10^−3^	3.1 × 10^−3^
*Cd*	0.010	0.0024	0.0032			2.7 × 10^−6^	3.7 × 10^−6^
*Hg*	0.131	0.0313	0.0420			NA ^a^	NA ^a^
*Pb*	0.005	0.0012	0.0016			3.0 × 10^−8^	4.1 × 10^−8^
**Fish from Bangladesh**							
*As*	1.800	0.5400	0.7333	0.5766	0.7830	2.4 × 10^−3^	3.3 × 10^−3^
*Cd*	0.082	0.0246	0.0334			2.8 × 10^−5^	2.8 × 10^−5^
*Hg*	0.017	0.0051	0.0069			NA ^a^	NA ^a^
*Pb*	0.023	0.0069	0.0093			1.8 × 10^−7^	2.3 × 10^−7^

**^a^** Value not available due to no known carcinogenic slope factor (CSFo) for mercury. Abbreviations: As: arsenic, Cd: cadmium, HI: hazard index, Hg: mercury, Pb: lead, THQ: target hazard quotient, TR: target carcinogenic risk, w.w.: wet weight.

## Data Availability

All data generated or analysed during this study are included in this published article.

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
