# Peer review of "Metal Contents in Fish from the Bay of Bengal and Potential Consumer Exposure—The EAF-Nansen Programme"

_foods, 2021, doi:10.3390/foods10051147_

Round 1

Reviewer 1 Report

Excellent contribution to the ongoing discussions on the risks and benefits of fish consumption. The paper is in particular important as it contributes with data on contaminants from species and areas where such data is very limited.

In my view, only a few minor questions and suggestions from my side:

Line 37-41: Although the focus of the paper is on metal content in fish, fish can be source of other contaminants as well due to pollution. Suggest amending this sentence to make this clear.

Table 2: Does footnote a refer to “Fish from Bangladesh” or “Number of fish in each composite sample” of Sardinella fimbriata?

Line 364: Suggest making it clear this sentence is about Cadmium and not MeHg.

Finally, in the discussion I would recommend to add a couple of more sentences on the benefits of consuming the fish. In addition to the minimal risk related to the metal content there are a number of benefits linked to fish consumption.

Reviewer 2 Report

The data shown in this manuscript is informative. There is only one comment.

Please clearly define the value of C (metal content in fish)in Equation 1  and Equation 3, it is the average value shown in Table 4 ?  If so, I would strongly suggest that authors recalculate these again, using upper value of 95% confidence limit, instead of average value, to ensure the actual safety. The reason is obvious, the metal content of half of the samples exceeds this value.
